# CGD: Modifying the Loss Landscape by Gradient Regularization

## Abstract

Line-search methods are commonly used to solve optimization problems. The simplest line search method is the steepest descent where we always move in the direction of the negative gradient. Newton's method on the other hand is a second-order method that uses the curvature information in the Hessian to pick the descent direction. In this work, we propose a new line-search method called Constrained Gradient Descent (CGD) that implicitly changes the landscape of the objective function for efficient optimization. CGD is formulated as a solution to the constrained version of the original problem where the constraint is on a function of the gradient. We optimize the corresponding Lagrangian function thereby favourably changing the landscape of the objective function. This results in a line search procedure where the Lagrangian penalty acts as a control over the descent direction and can therefore be used to iterate over points that have smaller gradient values, compared to iterates of vanilla steepest descent. We reinterpret and draw parallels with the Explicit Gradient Regularization (EGR) method, discussing its drawbacks and potential enhancements. Numerical experiments are conducted on synthetic test functions to illustrate the performance of CGD and its variants.

## 1 Introduction

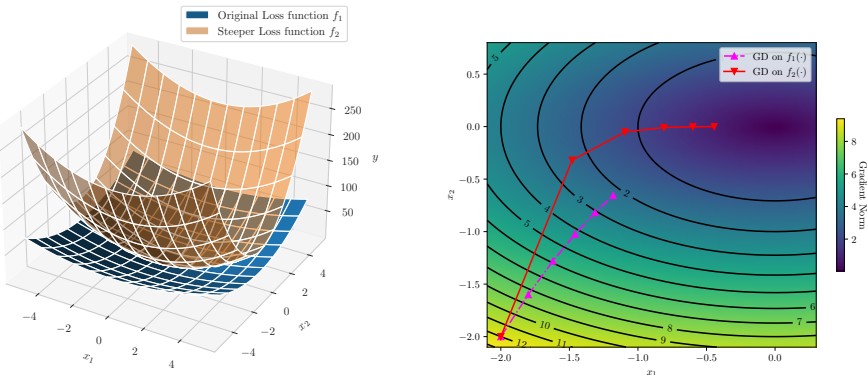

Figure 1: *(Left)* Loss function $f_1(\mathbf{x}) = x_1^2 + 2x_2^2$ and a steeper loss function $f_2(\mathbf{x})$. *(Right)* GD trajectories on functions $f_1(\mathbf{x})$ and $f_2(\mathbf{x})$. The two trajectories correspond to 5 iterations of GD on functions $f_1(\mathbf{x})$ and $f_2(\mathbf{x})$ respectively for fixed step size $\alpha = 0.05$. The contours and the gradient norm heatmap are over the function $f_1(\mathbf{x})$.

Deep Neural Networks have been successful at a variety of tasks primarily due to their over-parameterization. However, with such over-parameterization comes the problem of an uneven loss landscape that contains many local maxima and minima with some providing better performance than the others (Neyshabur et al., 2017; Li et al., 2018). As a result, optimizing over the parameter space has become incredibly relevant in the current times. To tackle this problem, one of the approach that is used is to modify the loss/objective function in such a way that the new function is easier to optimize. This is typically done by adding a regularization term that will make the

loss landscape more smooth, and which would in turn reduce the chances of the optimizer reaching sub-optimal minima, thereby improving generalizability. Considerable research in deep learning is now focused on designing loss functions that have smoother landscapes and therefore are easier to optimize (Du et al., 2022; Zhuang et al., 2022; Barrett & Dherin, 2022; Zhao et al., 2022; Karakida et al., 2023).

In this work, we focus on this idea of modifying the objective function for effective numerical optimization and propose an algorithmic paradigm to achieve it. To illustrate the key idea, consider minimizing a loss function $f_1(\cdot)$ over $\mathbb{R}^2$ as shown in Figure 1 *(Left)*. Additionally consider a steeper loss function $f_2(\cdot)$ such that for every $\mathbf{x} \in \mathbb{R}^2$ we have $f_1(\mathbf{x}) \leq f_2(\mathbf{x})$. Note that the functions are such that their minima coincides. Now perform gradient descent (GD) with a fixed step size on both these functions. Their respective trajectories are plotted in Figure 1 *(Right)* on the contours of $f_1$. It is clear from the figure that the iterates of GD on $f_2$ are much closer to the minima as compared to iterates of GD on $f_1$. We make this observation a focal point of this work and investigate if one can achieve the latter iterates (iterates from GD on $f_2$) on the former function ($f_1$).

Towards this, we propose Constraint gradient descent (CGD), a variant of the GD algorithm which achieves this by constraining the gradient norm to be appropriately small. Instead of solving this constrained optimization problem, we consider the Lagrangian of this function and perform GD on it. The Lagrangian parameter $\lambda$ controls the penalty on gradient norm and thereby controls the steepness of the modified function. It is on this modified function that we seek to apply GD to possibly attain iterates that are much closer to the local minima as compared to GD iterates. From Figure 1 *(Right)* we additionally see that compared to GD, the CGD iterates (which is nothing but GD iterates on $f_2$) have a lower gradient norm. Since the path to minima is over points with lower gradient norm, this we believe is an attractive feature and even amounts to better generalization properties for high dimensional functions such as loss landscapes of neural networks.

The idea of flat minima and their attractiveness goes long back to Hochreiter & Schmidhuber (1997): An optimal minimum is considered "flat" if the test error changes less in its neighbourhood. Keskar et al. (2017) and Chaudhari et al. (2017) observe better generalization results for neural networks at flat minima. Gradient regularization on the other hand has only been recently studied from a perspective of deep neural networks (DNNs) (Zhao et al., 2022; Karakida et al., 2023; Barrett & Dherin, 2022; Smith et al., 2021) where the square of the $L_2$ norm is penalized.

While the above methods only penaliize the squared $L_2$-norm of the gradient, we propose a more general approach where the penalty could be on any positive function of the gradient. This opens doors to investigating potential applications where penalty functions beyond squared $L_2$ norm might prove more beneficial. A key aim of this work is to understand this gradient regularization from the perspective of a numerical optimization algorithm and identify properties and features that may not have been obvious earlier. We believe CGD and its variants discussed in this work have the ability to find flatter minimas which may prove to be useful for training of neural networks.

We summarize our contributions below:

- We propose a new line-search procedure called Constraint Gradient Descent (CGD) that performs gradient descent on a gradient regularized loss function.

- While CGD requires the Hessian information, we also propose a first-order variant of CGD using finite-difference approximation of the Hessian called *CGD-FD*. We define appropriate stopping criteria in CGD-FD for settings where gradient computation can be expensive.

- We conduct experiments over synthetic test functions to compare the performance of CGD and its variants compared to standard line-search procedures.

- Our work also provides new insights to gradient regularization based methods. In fact, we re-interpret and identify pitfalls in Explicit Gradient Regularization (EGR) Methods using our formulation.

The rest of the paper is organized as follows. In the next section, we recall some preliminaries on line-search methods. We then discuss the CGD algorithm and propose its variants. We then illustrate the performance of our algorithm on several test functions and conclude with a discussion on future directions.

## 2 NOTATION AND PRELIMINARIES

The set of real numbers and non-negative real numbers is denoted by $\mathbb{R}$ and $\mathbb{R}^+$ respectively. We consider a vector $\mathbf{x} \in \mathbb{R}^n$ as a column vector given by $\mathbf{x} = [x_1 \quad x_2 \quad \ldots \quad x_n]^T$. We use a boldface letter to denote a vector and lowercase letters (with subscripts) to denote its components. $L^p$-norm of a vector $\mathbf{x}$ is defined as $\|\mathbf{x}\|_p = (\sum_i |x_i|^p)^{1/p}$. Setting $p = 2$ gives us $L^2$-norm: $\|\mathbf{x}\| \triangleq \|\mathbf{x}\|_2 = \sqrt{\sum_i x_i^2} = \sqrt{\mathbf{x}^T \mathbf{x}}$. $I_n$ denotes the identity matrix of size $n \times n$. All zero vector of length $n$ is denoted by $\mathbf{0}_n$.

### 2.1 LINE-SEARCH METHODS

Let $f(\mathbf{x})$ be a twice differential function with domain $\mathcal{D} \subseteq \mathbb{R}^n$ and codomain $\mathbb{R}$, i.e., $f : \mathcal{D} \to \mathbb{R}$. Let $\nabla f(\mathbf{x}_k)$ and $H(\mathbf{x}_k)$ denote the gradient and Hessian of the function $f(\mathbf{x})$ evaluated at point $\mathbf{x}_k \in \mathcal{D}$. For the sake of simplicity, we will also use the notation $\nabla f_k$ and $H_k$ to denote $\nabla f(\mathbf{x}_k)$ and $H(\mathbf{x}_k)$ respectively. For the given function $f(\mathbf{x})$, we consider the problem of finding its minimizer, i.e., we wish to find $\mathbf{x}^* \in \mathcal{D}$ such that

$$\mathbf{x}^* = \arg\min_{\mathbf{x} \in \mathcal{D}} f(\mathbf{x}). \tag{1}$$

We focus on the situation where $\mathbf{x}^*$ is obtained using an iterative line-search procedure (for details refer Nocedal & Wright (2006, Ch. 3)). The steepest descent (*gradient descent*) method, Newton's method, and quasi-Newton's method are some examples of line-search methods. In an iterative algorithm, the key idea is to begin with an initial guess $\mathbf{x}_0$ for the minimizer and generate a sequence of vectors $\mathbf{x}_0, \mathbf{x}_1, \ldots$ until convergence (or until desired level of accuracy is achieved). In such algorithms, $\mathbf{x}_{k+1}$ is obtained using $\mathbf{x}_k$ using a pre-defined update rule. For line-search methods, the update rule can be written in a general form as follows,

$$\mathbf{x}_{k+1} = \mathbf{x}_k + \alpha \mathbf{p}_k \tag{2}$$

where $\alpha \in \mathbb{R}^+$ is the step size (or learning rate) and $\mathbf{p}_k \in \mathbb{R}^n$ corresponds to the direction in the $k$-th iteration. For some line-search methods, we can express $\mathbf{p}_k$ as $\mathbf{p}_k = -P_k \nabla f_k$ where $P_k \in \mathbb{R}^{n \times n}$. For example, for steepest gradient descent method we have $P_k = -I_n$ and for Newton's method $P_k = -H_k^{-1}$. For $\mathbf{p}_k$ to be a *descent* direction, the following condition must hold:

$$\nabla f_k^T \mathbf{p}_k < 0. \tag{3}$$

## 3 CONSTRAINED GRADIENT DESCENT

In this section, we propose an iterative line-search method to find a minimizer $\mathbf{x}^*$ of the given function $f(\mathbf{x})$ (see Equation 1). We shall refer to our approach as *Constrained Gradient Descent* (CGD) method. It is known that if $\mathbf{x}^*$ is a local minimizer of the function $f(\mathbf{x})$ then the gradient at point $\mathbf{x}^*$ is equal to zero (Nocedal & Wright, 2006, Ch. 2). The key idea in our approach is to focus on the set of $\mathbf{x} \in \mathcal{D}$ such that $\nabla f(\mathbf{x})$ is close to zero. For this consider a general constrained optimization problem:

$$\mathbf{x}^* = \arg\min_{\substack{\mathbf{x} \in \mathcal{D} \\ h(\nabla f(\mathbf{x})) \le \epsilon}} f(\mathbf{x}) \tag{4}$$

where the constraint $h(\cdot)$ is defined on the gradient and $\epsilon$ is a small positive real number. The unconstrained optimization problem corresponding to Equation 4 is obtained by penalizing the gradient constraint with a Lagrange multiplier $\lambda > 0$:

$$\mathbf{x}^\star = \arg\min_{\mathbf{x} \in \mathcal{D}} \left[ f(\mathbf{x}) + \lambda \, h(\nabla f(\mathbf{x})) \right] \tag{5}$$

Note that $\epsilon$ doesn't affect the optimization and hence has been removed from the objective. Observe that such a formulation provides us with a *modified* loss function to optimize over. Choosing a non-negative constraint $h(\cdot)$ is preferred so that we do not introduce artificial minima points. In this case, the modified loss function will become steeper and hence easier to descent on than the original loss function. This can also be referred to by *penalization* or using a *gradient penalty* since the objective is penalized at points which are not local minima depending on the gradients at these points.

We consider non-decreasing functions over the gradient-norm to be suitable choices for the constraint function $h(\cdot)$ since the above mentioned qualities are fulfilled by them.

Consider, two modified loss functions using $L^p$-norm gradient penalty:

$$g_p(\mathbf{x}) \triangleq f(\mathbf{x}) + \lambda\|\nabla f(\mathbf{x})\|_p \qquad (L^p\text{-norm}) \qquad (6)$$

$$\hat{g}_p(\mathbf{x}) \triangleq f(\mathbf{x}) + \lambda\|\nabla f(\mathbf{x})\|_p^p \qquad (L^p\text{-norm to the power } p) \qquad (7)$$

For ease of exposition, we use $\hat{g}_2(\mathbf{x})$ for most of our discussion and numerical results. We simplify the notation further to denote $g(\mathbf{x}) \triangleq \hat{g}_2(\mathbf{x})$. More details on using functions $g_p(\mathbf{x})$ and $\hat{g}_p(\mathbf{x})$ can be found in Appendix A. Now consider the following optimization problem,

$$\mathbf{x}^\star = \arg\min_{\mathbf{x}\in\mathcal{D}} g(\mathbf{x}) \qquad (8)$$

where $g(\mathbf{x}) \triangleq f(\mathbf{x}) + \lambda\|\nabla f(\mathbf{x})\|^2 = f(\mathbf{x}) + \lambda\left(\nabla f(\mathbf{x})^T \nabla f(\mathbf{x})\right)$. We now apply steepest descent iteration to the optimization problem in Equation 8, given as

$$\begin{aligned}
\mathbf{x}_{k+1} &= \mathbf{x}_k - \alpha\nabla g_k \\
&= \mathbf{x}_k - \alpha\left(\nabla f_k + 2\lambda H_k \nabla f_k\right) \qquad (9) \\
&= \mathbf{x}_k - \alpha B_k \nabla f_k
\end{aligned}$$

where $B_k \triangleq I_n + 2\lambda H_k$. From Equation 2 note that we have $\mathbf{p}_k = -B_k\nabla f_k$ and $P_k = B_k$ which is the matrix corresponding to the direction taken at the $k$-th iteration by our CGD method.

Revisiting the example in Figure 1, the function $f_1(\mathbf{x}) = x_1^2 + 2x_2^2$ was penalized with the square of $L^2$-norm of gradient as given in Equation 8. Thus, the modified loss function $f_2(\mathbf{x}) = x_1^2 + 2x_2^2 + \lambda\left((2x_1)^2 + (4x_2)^2\right) = (1 + 4\lambda)x_1^2 + 2(1 + 8\lambda)x_2^2$ where $\lambda$ was chosen to be $0.4$.

We now provide a lemma that investigates if the penalized objective function has stationary points which are different from the original function and if so characterizes them.

**Lemma 1.** *Let $\mathcal{S}_{\hat{\mathbf{x}}}$ and $\mathcal{S}_{\mathbf{x}^*}$ be the stationary points of $f(\mathbf{x})$ and $g(\mathbf{x})$ as defined in Equation 8. Then, $\mathcal{S}_{\hat{\mathbf{x}}} \subseteq \mathcal{S}_{\mathbf{x}^*}$. Furthermore, for any $\mathbf{x}^* \in \mathcal{S}_{\mathbf{x}^*}$ one of the following is true: (a) $\mathbf{x}^* \in \mathcal{S}_{\hat{\mathbf{x}}}$ or (b) $\nabla f(\mathbf{x}^*)$ is an eigenvector of $H(\mathbf{x}^*)$ with the eigenvalue $-\frac{1}{2\lambda}$.*

*Proof.* For any $\hat{\mathbf{x}} \in \mathcal{S}_{\hat{\mathbf{x}}}$ we have $\nabla f(\hat{\mathbf{x}}) = \mathbf{0}_n$ and hence $\nabla g(\hat{\mathbf{x}}) = \mathbf{0}_n$ from Equation 9. Thus, $\mathcal{S}_{\hat{\mathbf{x}}} \subseteq \mathcal{S}_{\mathbf{x}^*}$ trivially holds. Now for any $\mathbf{x}^* \in \mathcal{S}_{\mathbf{x}^*}$,

$$\nabla g(\mathbf{x}^*) = 0 \implies (I_n + 2\lambda H(\mathbf{x}^*))\nabla f(\mathbf{x}^*) = 0$$

Therefore if $\nabla f(\mathbf{x}^*) = 0$, we have $\mathbf{x}^* \in \mathcal{S}_{\hat{\mathbf{x}}}$. Otherwise, $H(\mathbf{x}^*)\nabla f(\mathbf{x}^*) = -\frac{1}{2\lambda}\nabla f(\mathbf{x}^*)$. $\qquad\square$

The above lemma illustrates that additional stationary points could possibly be introduced and some of these points could also be a local minima. The above lemma also characterizes conditions under which this is true and therefore such points can easily be detected. A perturbation from the current $\lambda$ in that case results in the iterate to descend further, possibly moving towards a better local minima.

Within the current scheme, the nature of stationary points $f(\mathbf{x})$ might change in $g(\mathbf{x})$. Particularly, the local maxima and saddle points of $f(\mathbf{x})$ might become local minima in $g(\mathbf{x})$ depending on the coefficient $\lambda$. This, happens due to the fact that at the stationary points, the modified function takes the value of the original function itself (since gradient is zero). But within the neighbourhood, the function value increases (since we are adding gradient-norm at these points). Thus, for a particular local maximum (or saddle point) $\mathbf{y}$, the nature of the point changes to that of a local minimum when $\lambda > \lambda^*$ where $\lambda^*$ represents a threshold for this behaviour. $\lambda^*$ depends on the function's curvature and gradient norm changes in the neighbourhood of the point $\mathbf{y}$ and hence, is different for different local maxima and saddle points.

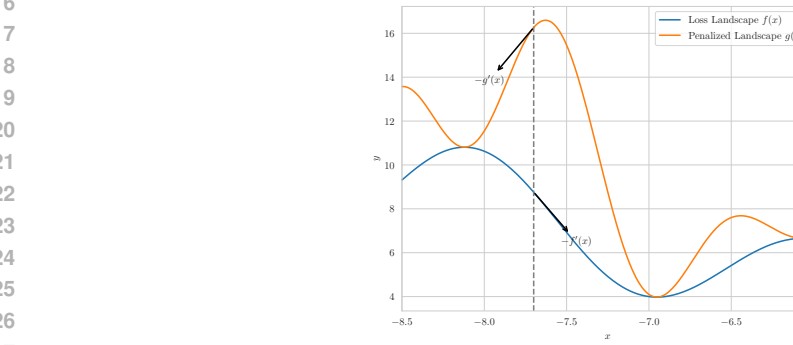

Figure 2: For the given function, observe artificial stationary points being introduced after penalization at $x \approx -6.5$ and $x \approx -7.6$. Both these stationary points turn out to be local maxima. Observe how the local maximum at $x \approx -8.1$ turns into a local minimum. We also plot the descent directions $-f'(x)$ and $-g'(x)$ at $x = -7.7$ (denoted by dotted line).

Thus within the current scheme, descent on the penalized function might actually cause ascent on the original one (moving towards the local maxima turned into local minima). For example, in Figure 2 we observe that at $x = -7.7$, steepest descent along $-\nabla g(\mathbf{x})$ direction would actually cause an ascent over the original loss function.

To fix this behaviour, we ensure that we only move along the direction $-\nabla g(\mathbf{x})$ when it is a descent direction. Therefore Equation 3 requires that $\nabla f(\mathbf{x})^T \nabla g(\mathbf{x}) > 0$ in order for us to move along $-\nabla g(\mathbf{x})$. If Equation 3 doesn't hold i.e, $\nabla f(\mathbf{x})^T \nabla g(\mathbf{x}) \geq 0$ then we set $\lambda = 0$ at this point and only move along $-\nabla f(\mathbf{x})$ direction. Note that this simple check also helps us to avoid stopping at artificially introduced stationary points. We summarize CGD in Algorithm 1.

---

**Algorithm 1** Constrained Gradient Descent (CGD)

---

**Input**: Objective function $f : \mathcal{D} \to \mathbb{R}$, initial point $\mathbf{x}_0$, max iterations $T$, step size $\alpha$, regularization coefficient $\lambda$.
**Output**: Final point $\mathbf{x}_T$.

1: **for** iteration $k = 0, \ldots, T - 1$ **do**
2:     $\mathbf{p}_k \leftarrow -\left(I_n + 2\lambda_k H_k\right)\nabla f_k$
3:     **if** $\nabla f_k^T \mathbf{p}_k < 0$ **then**                           ▷ Check if $\mathbf{p}_k$ is a Descent Direction
4:         $\mathbf{x}_{k+1} \leftarrow \mathbf{x}_k + \alpha\mathbf{p}_k$
5:     **else**
6:         $\mathbf{x}_{k+1} \leftarrow \mathbf{x}_k - \alpha\nabla f_k$
7:     **end if**
8: **end for**
9: **return** $\mathbf{x}_T$

---

### 3.1 CGD-FD: Finite difference approximation of the Hessian

Note that CGD is a second-order optimization algorithm as it requires the Hessian information to compute each iterate. We improve over this complexity by restricting CGD to be a first-order line search method wherein the Hessian is approximated using a finite difference (Pearlmutter, 1994). Using Taylor series, we know that

$$\nabla f(\mathbf{x}_k + \Delta\mathbf{x}) = \nabla f_k + H_k\Delta\mathbf{x} + \mathcal{O}(\|\Delta\mathbf{x}\|^2).$$

Now let $\Delta\mathbf{x} = r\mathbf{v}$ where $r$ is arbitrarily small. Then, we can rewrite the above expression as:

$$H\mathbf{v} = \frac{\nabla f(\mathbf{x}_k + r\mathbf{v}) - \nabla f_k}{r} + \mathcal{O}(\|r\|).$$

Substituting $\mathbf{v} = \nabla f_k$ and using this approximation in Equation 9 gives us

$$\nabla g(\mathbf{x}_k) \approx \nabla f_k + 2\lambda \frac{\nabla f(\mathbf{x}_k + r\nabla f_k) - \nabla f_k}{r}$$
$$= (1 - \nu)\nabla f_k + \nu\nabla f(\mathbf{x}_k + r\nabla f_k) \tag{10}$$

where $\nu = 2\lambda/r$. We call this version *CGD with Finite Differences* (CGD-FD).

Note now that we require two gradient calls in every iteration of CGD-FD which may be prohibitive. Towards this it is only natural to consider a stopping criteria so as to revert back to using steepest direction whenever the improvment through CGD-FD iterates is low. This is done to avoid the extra gradient evaluations after the optimizer has moved to a sufficiently optimal point in the domain. Such criteria are particularly helpful in loss functions where gradient evaluations are costly and therefore, budgeted. This also gives us the freedom to play around with which direction to choose (steepest or CGD-FD) and quantify when to switch to the other. For instance, we intially only move in the directions suggested by CGD-FD and switch to steepest once we make good-enough drop in the function value from the initial point.

In our experiments, we used a combination of the strategies explained above. We only use CGD-FD for the first $b$ iterations (out of the total budget $T$) to strive for a good-drop in function values initially. While doing so, we also ensure that we only move in the direction $\mathbf{p}_k$ (CGD-FD, Equation 10) if it is a descent direction. Otherwise, we stop using CGD-FD updates henceforth and use steepest direction from here onwards. We summarize our CGD-FD method in Algorithm 2.

---

**Algorithm 2** Constrained Gradient Descent using Finite Differences (CGD-FD)

---

**Input**: Objective function $f : \mathcal{D} \to \mathbb{R}$, initial point $\mathbf{x}_0$, max iterations $T$, step size $\alpha$, regularization coefficients $\lambda$, stopping threshold $b$.
**Output**: Final point $\mathbf{x}_T$.

1: $\nu \leftarrow 2\lambda/r$
2: USE_CGD $\leftarrow$ TRUE
3: Number of Gradient Evaluations $c \leftarrow 0$                ▷ Can be interpreted as cost
4: **for** $k = 0, \ldots, T - 1$ **do**
5:      **if** $c = T$ **then return** $\mathbf{x}_k$                ▷ Budget Exhausted
6:      **end if**
7:      **if** USE_CGD **then**
8:          $\mathbf{p}_k \leftarrow -[(1 - \nu)\nabla f_k + \nu\nabla f(\mathbf{x}_k + r\nabla f_k)]$
9:          $c \leftarrow c + 2$
10:          **if** $\nabla f_k^T \mathbf{p}_k < 0$ **then**            ▷ Check if $\mathbf{p}_k$ is a Descent Direction
11:             $\mathbf{x}_{k+1} \leftarrow \mathbf{x}_k + \alpha\mathbf{p}_k$
12:          **else**
13:             $\mathbf{x}_{k+1} \leftarrow \mathbf{x}_k - \alpha\nabla f_k$
14:             USE_CGD $\leftarrow$ FALSE             ▷ Stop using CGD-FD iterates
15:          **end if**
16:      **else**
17:          $\mathbf{x}_{k+1} \leftarrow \mathbf{x}_k - \alpha\nabla f_k$
18:          $c \leftarrow c + 1$
19:      **end if**
20:      **if** $k \geq b$ **then**
21:          USE_CGD $\leftarrow$ FALSE
22:      **end if**
23: **end for**
24: **return** $\mathbf{x}_T$

---

## 4 NUMERICAL EXPERIMENTS

We test CGD-FD on synthetic test functions from *Virtual Library of Simulation Experiments: Test Functions and Datasets* [1].

---

[1] http://www.sfu.ca/~ssurjano

For each function, we strive for an initial gain i.e., a good-drop in function value while using CGD-FD iterates. We set the stopping threshold $b$ as $T/4$ where $T$ is the total budget. For our experiments we chose $T = 40$.

For the hyperparameter $\lambda$, we empirically tested different schedules and strategies for different kinds of functions. We observe that using constant $\lambda$ works very well with convex functions. However, in non-convex functions an increasing schedule might be useful. The intuition behind an increasing schedule is that initially we want to penalize less and as we move to points with lower gradient norm, we can start to penalize more. We denote this as $\text{Linear}(a, b)$ which is an increasing linear schedule of $T$ values going from $a$ to $b$. For the step-size $\alpha$, a constant-value was found to be suitable in all our experiments.

To measure the initial drop in function value, we compare the *Improvement* for the first step of CGD-FD vs the first step of steepest descent. Improvement in the first step is defined as:

$$\text{Improvement (in \%)} = \frac{f(\mathbf{x}_0) - f(\mathbf{x}_1)}{f(\mathbf{x}_0)} * 100$$

Table 1 summarizes the hyperparameter values chosen and Table 2 compares the intial improvement across CGD-FD and steepest descent.

Table 1: Choices of $\alpha$ and $\lambda$ for synthetic test functions (Budget $T = 40$, Stopping threshold $b = T/4$).

| Test Function (Dimensions=$n$) | $\lambda$ | $\alpha$ |
|---|---|---|
| Quadratic function ($n = 10$) | 0.4 | 0.01 |
| Rotated hyper-ellipsoid function ($n = 5$) | 0.5 | 0.01 |
| Levy function ($n = 2$) | Linear(0.01, 0.1) | 0.05 |
| Branin function ($n = 2$) | 0.07 | 0.01 |
| Griewank function ($n = 2$) | 40.0 | 0.01 |
| Matyas function ($n = 2$) | 10.0 | 0.01 |

Table 2: Initial Improvement (in %) across CGD-FD and Steepest Descent (GD).

| Test Function (Dimensions=$n$) | GD | CGD-FD |
|---|---|---|
| Quadratic function ($n = 10$) | 18.89 | **97.91** |
| Rotated hyper-ellipsoid function ($n = 5$) | 15.94 | **82.76** |
| Levy function ($n = 2$) | 23.73 | **63.21** |
| Branin function ($n = 2$) | 37.53 | **87.07** |
| Griewank function ($n = 2$) | 0.01 | **0.08** |
| Matyas function ($n = 2$) | 1.83 | **34.40** |

We plot the function values vs gradient evaluations in Figure 3. Note, that the $x$-axis is not iterations but gradient evaluations. We compare the trajectories across points where equal number of gradients have been evaluated.

## 5 EXPLICIT GRADIENT REGULARIZATION (EGR)

Barrett & Dherin (2022) proposed **Explicit Gradient Regularization** (EGR) where the original loss function is regularized with the square of $L^2$-norm of the gradient. This regularized objective is then optimized with the intention that the model's parameters will converge to a *flat*-minima and thus, be more generalizable. However, an understanding of why this happens is missing.

We explain EGR through the example visualized in Figure 4 *(Left)*. Based on our formulation explained in Section 3, we can interpret the gradient-regularized function as a more steeper version of the orginal loss function wherein the minima remain same. Due to the gradient penalty, a steeper minima (in the original loss function) would turn more steeper while the increase in steepeness

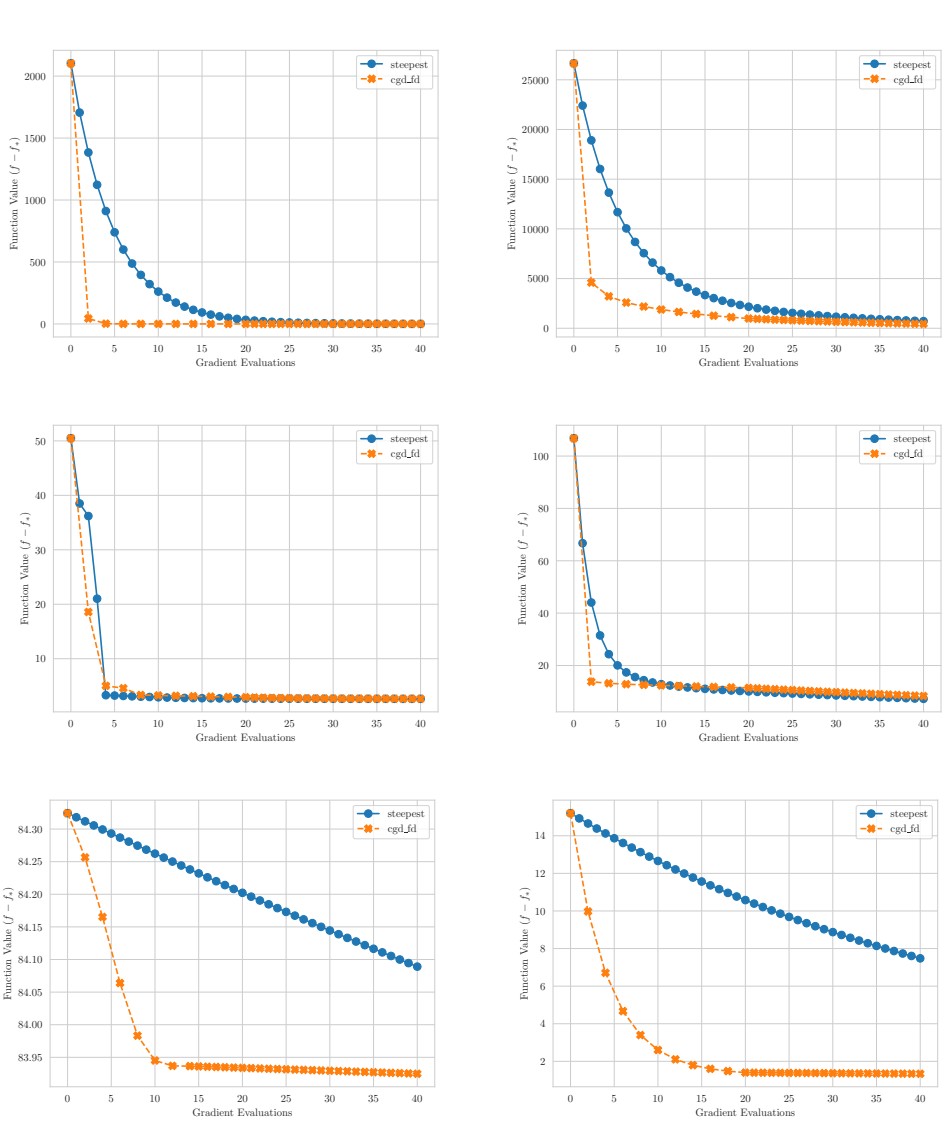

Figure 3: Function value $f(\cdot) - f^*$ ($f^*$ being the optimal function value) vs Gradient Evaluations. In raster order (from top to bottom; left to right), the functions are: Quadratic function, Rotated hyper-ellipsoid function, Levy function, Branin function, Griewank function and Matyas function (see Appendix B). *Note*: The $x$-axis of each plot is not iterations but number of gradients evaluated.

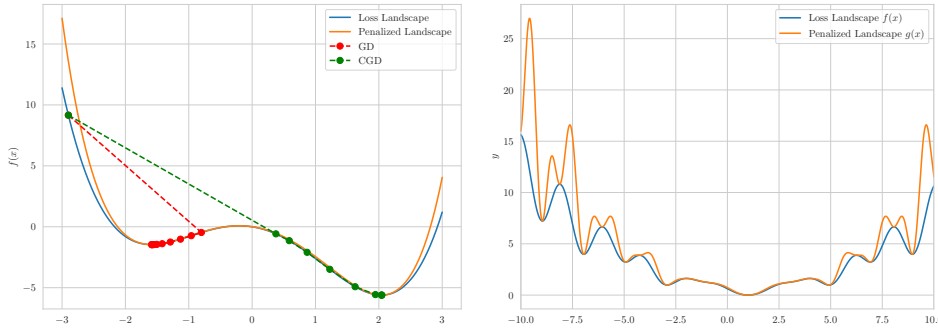

Figure 4: *(Left)* CGD is able to move to a better local minimum by moving along the negative gradients over the modified loss function. *(Right)* Local maxima of the original loss function turning into local minima of the penalized loss function.

wouldn't be this high at a flatter minimum. Thus, at a constant step-size, the optimizer will likely overshoot over the sharper minimum due to the extremely high gradient value, while still being able to converge to the less-steeper and preferred flat minimum point. In Figure 4 *(Left)* we see how GD gets stuck at a suboptimal point of the function while CGD (GD over the penalized loss function) is able to avoid this point and converges to a better (more optimal) minimum point.

EGR has the drawbacks of having fictitious minima identified in Lemma 1. Since, the loss landscape of neural networks is highly uneven (Li et al., 2018), it's likely that artificial stationary points are introduced and the optimizer might converge at local-maxima points since their nature changes with regularization. For example, in Figure 4 *(Right)* we observe artificial stationary points being introduced between local-minima and local-maxima of the original loss function while some local-maxima also turn into local-minima of the penalized loss function. Therefore, one needs to ensure that there are suitable fixes for these scenarios for better performance. Specifically, the direction along of the penalized loss function, $-\nabla g(\cdot)$ should be a descent direction and that we shouldn't stop at points where $\nabla g(\cdot) = \mathbf{0}$ but $\nabla f(\cdot) \neq \mathbf{0}$ as discussed in Lemma 1.

Another downfall with EGR is that it requires hessian evaluation for each mini-batch while training. This might become a costly operation for bigger models and hence some approximation of the hessian should be used. Another thing we observe from the experimental results for CGD-FD, is that the improvment through optimizing the regularized function is mostly only during the initial steps. Hence, one should only use EGR for some initial steps to reach a good-enough starting point while not exhausting the budget for gradient evaluations.

## 6 FUTURE WORKS

In this work, we considered a new line swarch method that penalizes the norm of the gradient and provides iterates that have lower gradient norm compared to vanilla gradient descent. We identify properties of this algorithm, provide a varient that does not require Hessian and illustrate connections to the widely popular explicit gradient regularization literature.

There are several future directions arising from this work. We would like to investigate in greater detail the role of different penalty functions and norms beyond $L_2$. We would like to investigate applications of this method in more diverse settings like reinforcement learning and even Bayesian optimization. We hope this work sputters more discussions on gradient regularization helping in neural network generalization.

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

## A $L^p$-NORM GRADIENT PENALTY

**Lemma 2.** *For the modified functions $g_p(\mathbf{x})$ and $\hat{g}_p(\mathbf{x})$ defined in Equations 6 and 7, the gradients are given as follows:*

$$\nabla g_p(\mathbf{x}) = \nabla f(\mathbf{x}) + \frac{\lambda}{\|\nabla f(\mathbf{x})\|_p^{p-1}} H(\mathbf{x}) \left( \nabla f(\mathbf{x}) \odot \mathrm{abs}\left( \nabla f(\mathbf{x}) \right)^{p-2} \right)$$

$$\nabla \hat{g}_p(\mathbf{x}) = \nabla f(\mathbf{x}) + \lambda p H(\mathbf{x}) \left( \nabla f(\mathbf{x}) \odot \mathrm{abs}\left( \nabla f(\mathbf{x}) \right)^{p-2} \right)$$

*where $\odot$ is element-wise product of vectors and $\mathrm{abs}$ is element-wise absolute $|\cdot|$ of the vector.*

*Proof.* We know that $L^p$-norm of $\mathbf{x}$ is $\|\mathbf{x}\|_p = \left( \sum_i |x_i|^p \right)^{1/p}$. We first find $\nabla \|\mathbf{x}\|_p$. For some $x_k$:

$$\frac{\partial \|\mathbf{x}\|_p}{\partial x_k} = \frac{1}{p} \left( \sum_i |x_i|^p \right)^{\frac{1}{p}-1} \frac{\partial}{\partial x_k} \left\{ \sum_i |x_i|^p \right\}$$

$$= \frac{1}{p} \left( \sum_i |x_i|^p \right)^{\frac{1}{p}-1} \left( \sum_i \frac{\partial |x_i|^p}{\partial x_k} \right)$$

$$= \frac{1}{p} \left( \sum_i |x_i|^p \right)^{\frac{1}{p}-1} \left( \sum_i p|x_i|^{p-1} \frac{\partial |x_i|}{\partial x_k} \right)$$

$$= \left( \sum_i |x_i|^p \right)^{\frac{1}{p}-1} \left( \sum_i |x_i|^{p-1} \frac{x_i}{|x_i|} \frac{\partial x_i}{\partial x_k} \right)$$

$$= \left( \sum_i |x_i|^p \right)^{\frac{1-p}{p}} \left( \sum_i x_i |x_i|^{p-2} \frac{\partial x_i}{\partial x_k} \right)$$

$$= \|\mathbf{x}\|_p^{1-p} \left( \sum_i x_i |x_i|^{p-2} \frac{\partial x_i}{\partial x_k} \right) \text{ or } \frac{1}{\|\mathbf{x}\|_p^{p-1}} \left( \sum_i x_i |x_i|^{p-2} \frac{\partial x_i}{\partial x_k} \right)$$

Using this result we can obtain $\nabla \|\nabla f(\mathbf{x})\|_p$ by chain rule. Note that $\nabla f \triangleq \nabla f(\mathbf{x})$ in the following expressions:

$$\nabla \|\nabla f\|_p = \frac{1}{\|\nabla f\|_p^{p-1}} \begin{bmatrix} \sum_i \frac{\partial f}{\partial x_i} \left| \frac{\partial f}{\partial x_i} \right|^{p-2} \frac{\partial^2 f}{\partial x_i \partial x_1} \\ \vdots \\ \sum_i \frac{\partial f}{\partial x_i} \left| \frac{\partial f}{\partial x_i} \right|^{p-2} \frac{\partial^2 f}{\partial x_i \partial x_n} \end{bmatrix}$$

$$= \frac{1}{\|\nabla f\|_p^{p-1}} \begin{bmatrix} \frac{\partial^2 f}{\partial x_1^2} & \frac{\partial^2 f}{\partial x_1 \partial x_2} & \cdots & \frac{\partial^2 f}{\partial x_1 \partial x_n} \\ \vdots & & \vdots & \\ \frac{\partial^2 f}{\partial x_1 \partial x_n} & \frac{\partial^2 f}{\partial x_2 \partial x_n} & \cdots & \frac{\partial^2 f}{\partial x_n^2} \end{bmatrix} \begin{bmatrix} \frac{\partial f}{\partial x_1} \left| \frac{\partial f}{\partial x_1} \right|^{p-2} \\ \vdots \\ \frac{\partial f}{\partial x_n} \left| \frac{\partial f}{\partial x_n} \right|^{p-2} \end{bmatrix}$$

$$\implies \nabla \|\nabla f(\mathbf{x})\|_p = \frac{1}{\|\nabla f\|_p^{p-1}} H(\mathbf{x}) \left( \nabla f(\mathbf{x}) \odot \mathrm{abs}\left( \nabla f(\mathbf{x}) \right)^{p-2} \right) \tag{11}$$

We use the expression obtained in Equation 11 in computing derivative of $g_p(\mathbf{x})$ and $\hat{g}_p(\mathbf{x})$ (through chain-rule) as follows:

$$\nabla g_p(\mathbf{x}) = \nabla f(\mathbf{x}) + \lambda \nabla \|\nabla f(\mathbf{x})\|_p, \text{ and}$$

$$\nabla \hat{g}_p(\mathbf{x}) = \nabla f(\mathbf{x}) + \lambda p \|\nabla f(\mathbf{x})\|_p^{p-1} \nabla \|\nabla f(\mathbf{x})\|_p.$$

which on expansion proves the lemma. $\qquad\square$

**Corollary 1** (CGD with $L^1$-norm). *Consider modified loss functions $g_1(\mathbf{x})$ and $\hat{g}_1(\mathbf{x})$ (as stated in Equations 6 and 7) that use $L^1$-norm gradient penalty. Then,*

$$\nabla g_1(\mathbf{x}) = \nabla \hat{g}_1(\mathbf{x}) = \nabla f(\mathbf{x}) + \lambda H(\mathbf{x}) \operatorname{sign}(\nabla f(\mathbf{x}))$$

**Corollary 2** (CGD with $L^2$-norm). *Similarly, $g_2(\mathbf{x})$ and $\hat{g}_2(\mathbf{x})$ with $L^2$-norm gradient penalty have the following gradients:*

$$\nabla g_2(\mathbf{x}) = \nabla f(\mathbf{x}) + \frac{\lambda}{\|\nabla f(\mathbf{x})\|} H(\mathbf{x}) \nabla f(\mathbf{x})$$

$$\nabla \hat{g}_2(\mathbf{x}) = \nabla f(\mathbf{x}) + 2\lambda H(\mathbf{x}) \nabla f(\mathbf{x})$$

## B    EXPERIMENTAL RESULTS

The functions used for the experiments are given as follows:

- Quadratic Function (dimensions $n$)

$$f(\mathbf{x}) = \frac{1}{2}\mathbf{x}^T A \mathbf{x} - \mathbf{b}^T \mathbf{x} + c$$

  where $A$ is a positive-definite matrix, $\mathbf{b}$ is any vector. $\mathbf{x}^* = A^{-1}\mathbf{b}$ and $f^* = f(\mathbf{x}^*)$. In our experiments, we generate $A$ randomly as follows:

$$A = \frac{1}{2}(U + U^T) + nI_n$$

  where $U \sim \mathcal{U}[0, 1]^{n \times n}$, a matrix where all values are uniformly sampled from the interval $[0, 1]$. We take $\mathbf{b} = [1,\ 2,\ \ldots,\ n]^T$ and $c = 0.5$.

- Rotated hyper-ellipsoid Function (dimensions $n$)

$$f(\mathbf{x}) = \sum_{i=1}^{n} \sum_{j=1}^{i} x_j^2$$

  $f^* = 0$ at $\mathbf{x}^* = \mathbf{0}_n$.

- Matyas Function (dimensions $n = 2$)

$$f(\mathbf{x}) = 0.26(x_1^2 + x_2^2) - 0.48x_1 x_2$$

  $f^* = 0$ at $x^* = \mathbf{0}_n$.

- Branin Function (dimensions $n = 2$)

$$f(\mathbf{x}) = \left(x_2 - \frac{5.1}{4\pi^2}x_1^2 + \frac{5}{\pi}x_1 - 6\right)^2 + 10\left(1 - \frac{1}{8\pi}\right)\cos x_1 + 10$$

  $f^* = 0.397887$ at $\mathbf{x}^* = [-\pi, 12.275]^T, [\pi, 2.275]^T, [9.42478, 2.475]^T$.

- Levy Function (dimensions $n$)

$$f(\mathbf{x}) = \sin^2(\pi w_1) + \sum_{i=1}^{n-1}(w_i - 1)^2(1 + 10\sin^2(\pi w_i + 1)) + (w_n - 1)^2(1 + \sin^2(2\pi w_n))$$

  where $w_i = 1 + (x_i - 1)/4$ for all $i$. $f^* = 0$ at $\mathbf{x}^* = (1, \ldots, 1)$.

- Griewank Function (dimensions $n$)

$$f(\mathbf{x}) = \sum_{i=1}^{n} \frac{x_i^2}{4000} - \prod_{i=1}^{n} \cos\left(\frac{x_i}{\sqrt{i}}\right) + 1$$

  $f^* = 0$ at $x^* = \mathbf{0}_n$.

