# OpenReview forum: "CGD: Modifying the Loss Landscape by Gradient Regularization"
_ICLR.cc/2025/Conference — ICLR 2025 Conference Withdrawn Submission_

### Official Review · Reviewer_tBPs · 2024-10-25

**Soundness:** 2
**Presentation:** 2
**Contribution:** 1
**Rating:** 1
**Confidence:** 4

**Summary:**

The paper introduces Constrained Gradient Descent (CGD), an optimization technique that modifies the loss landscape by imposing constraints on the gradient's norm. A first-order variant, CGD-FD, is proposed by using finite-difference approximations of the Hessian to avoid computational cost. The work also highlights limitations of existing Explicit Gradient Regularization (EGR) methods.

**Strengths:**

The motivation is clear with nice visualization.

**Weaknesses:**

1. The introduction uses deep learning as a motivation but it should be others since the work is focusing on deterministic optimization.
2. lacks literature review many other related algorithms.
3. The contribution lacks novelty. The algorithm basically does the following: when the normalized Newton step is a descent direction, use that; Otherwise, just employ gradient descent. When $\lambda$ is sufficiently small, it is guaranteed to employ a normalized Newton step.
4. The authors misunderstands the line search method. Line search should be an adaptive step size scheme but the algorithm doesn't have such property.
5. The algorithm introduces another hyperparameter $\lambda$ to the algorithm. Compared to gradient descent, the tuning effort for CGD is greater.
6. lacks convergence result
7. It's not enough for comparing the algorithms over 6 problems, and it is not fair to tune  $\lambda$ for the CGD algorithm in the comparison with GD.

**Questions:**

See weaknesses

---

### Official Review · Reviewer_yHLq · 2024-10-29

**Soundness:** 1
**Presentation:** 1
**Contribution:** 1
**Rating:** 1
**Confidence:** 5

**Summary:**

This paper proposes a gradient norm penalty to the original objective function and follows the gradient of the penalized objective, resulting in multiplying (I + lambda * Hessian) to the gradient step. This modification basically does not change the global optimum of the original optimization problem. While the original proposal requires the Hessian computation, its finite difference approximation is also proposed. The performance is compared to the standard GD on 6 test problems up to 10 dimensions.

**Strengths:**

A simple approach to accelerate the gradient descent.

**Weaknesses:**

Evaluation. The proposed approach is only compared with a naive GD with a constant step-size. It is definitely not enough to show the advantages of the proposed approach. It should be compared with momentum-based approaches such as NAG, quasi-Newton approaches such as BFGS and L-BFGS, and conjugate gradient methods. For the line search part, comparison with other line search methods based on Armijo condition or Wolf condition should be performed. Comparison with some commercial software such as Matlab optimization toolbox as a baseline is also helpful to show the advantage.

The search space dimensionality is also limited. Though the authors mention about DNN  at the beginning of this paper, the performance was tested only on 6 test problems up to 10 dimensions. Using benchmarking testbed with wider coverage such as CUTEst is recommended.

No theoretical justification is given. I am curious to know whether this approach improves the convergence rate.

**Questions:**

See the weaknesses above.

---

### Official Review · Reviewer_edUM · 2024-10-30

**Soundness:** 2
**Presentation:** 2
**Contribution:** 2
**Rating:** 3
**Confidence:** 3

**Summary:**

This paper proposes Constraint Gradient Descent (CGD) and its first-order variant CGD-FD, which use gradient regularization and finite-difference approximations for efficient optimization, and compare their performance against standard methods, while also re-evaluating Explicit Gradient Regularization techniques.

**Strengths:**

This paper study an important research issue and this paper is generally well written.

**Weaknesses:**

see questions.

**Questions:**

1. I think the major issue is that, for such a theoretical paper, the theoretical results are just too weak. This is just a simple Lemma for that.

2. Besides, the proposed constrained strategy seems very easy to think of, is it really novel?

3. What is the difference with mirror gradient descent?

---

### Official Review · Reviewer_TPeE · 2024-10-30

**Soundness:** 2
**Presentation:** 2
**Contribution:** 1
**Rating:** 3
**Confidence:** 4

**Summary:**

This paper introduces Constrained Gradient Descent (CGD), a new line-search method that alters the objective function landscape for better optimization. CGD is based on a constrained version of the problem, optimizing the Lagrangian to control descent direction, potentially targeting points with smaller gradients than steepest descent. The authors relate CGD to Explicit Gradient Regularization (EGR), discussing its pros and cons, and validate CGD's performance through numerical tests on synthetic functions.

**Strengths:**

The paper is well written, and the method is well described.

**Weaknesses:**

* The statement in Section 3, especially the Lemma 1, is quiet confusing. Lemma 1 shows that the set of minimizers for the problem you defined in Equation (8) includes the set of minimizers for the original problem, which raises a convergence issue: where exactly will your defined algorithm converge to? The final convergence result might be inferior to the original gradient descent. This is also reflected in the results shown in Figure 2. Even with corrections made in the Algorithm 1, the convergence result is still not guaranteed.

*  Your algorithm 2, CGD-FD, is quiet similar with traditional nesterov momentum gradient descent, except that the momentum term in the Nesterov algorithm has been replaced with the gradient at the current point. However, this article does not compare with any momentum-based methods, and it is not necessarily superior to these methods, because the Nesterov algorithm theoretically has a convergence rate of $O(1/k^2)$.

**Questions:**

* How does your method compare in terms of effectiveness with momentum-based methods (such as the Nesterov method)?

---

### Note · Authors · 2024-11-17

**Comment:**

After careful consideration, we have decided to withdraw our submission from the conference. We sincerely appreciate the thoughtful feedback provided by the reviewers, which will help us refine and strengthen our work.

**Withdrawal Confirmation:**

I have read and agree with the venue's withdrawal policy on behalf of myself and my co-authors.